# Black–White Disparities in Criminal Justice Referrals to Drug Treatment: Addressing Treatment Need or Expanding the Diagnostic Net?

**DOI:** 10.3390/bs6040021

**Published:** 2016-10-02

**Authors:** Karen McElrath, Angela Taylor, Kimberly K. Tran

**Affiliations:** 1Department of Criminal Justice, Fayetteville State University, 1200 Murchison Road, Fayetteville, NC 28301, USA; ataylo14@uncfsu.edu; 2Department of Psychology, Fayetteville State University, 1200 Murchison Road, Fayetteville, NC 28301, USA; ktran@uncfsu.edu

**Keywords:** criminal justice referrals, treatment, marijuana use, race, African-American

## Abstract

Slightly more than half of admissions to U.S. publicly-funded treatment for marijuana use are referred by the criminal justice system; this pattern has remained for at least 20 years. Nationally, Blacks comprise nearly a third of treatment admissions for marijuana use. This article explores the interplay between race and criminal justice referrals to treatment for marijuana use. Using data from the (U.S.) 2011 Treatment Episode Data Set, we examine the relationship between race and diagnosis of cannabis use disorder (dependence versus abuse) among referrals to community-based treatment in North Carolina. We compare Black/White differences in cannabis diagnoses across four referral sources: the criminal justice system, healthcare providers, self, and other sources. Race was significantly related to type of diagnosis across all four referral sources, however, the nature of the relationship was distinctly different among criminal justice referrals with Whites being more likely than Blacks to be diagnosed with cannabis dependence. Moreover, the marijuana use profiles of criminal justice referrals differed substantially from individuals referred by other sources. The findings suggest that diagnoses of cannabis abuse (rather than dependence) may have worked to widen the diagnostic net by “capturing” individuals under control of the criminal justice system who manifested few problems with marijuana use, other than their involvement in the criminal justice system. The potential for a net-widening effect appeared to be most pronounced for Blacks.

## 1. Introduction

In the United States, individuals under control of the criminal justice system have experienced relatively high rates of current, persistent, or problem drug use. This pattern is well-established and extends to those who have been recently arrested [1,2], detained in jail [3,4], or sentenced to probation or prison [5]. Despite their excessive use of illicit substances, only a small proportion of arrestees, probationers, or prisoners are exposed to drug treatment in the criminal justice system [6]. Among individuals under community supervision, fewer than 10% who are in need of drug/alcohol services actually receive it [7]. Still, community supervision that encompasses quasi-coerced treatment has expanded [8], whereby participation in drug treatment occurs as an alternative to incarceration or as a requirement of probation or parole [9]. Under the guise of therapeutic jurisprudence [10,11], drug treatment is intertwined with punishment and social control, although the latter tends to take precedence over rehabilitation [12].

The estimated gap between treatment need and provision among criminal justice populations is of particular interest when considered in the context of admissions to publicly-funded substance use treatment. The criminal justice system is the second largest referral source to publicly-funded treatment in the U.S., accounting for 34% of the nearly 1.7 million admissions in 2013 [13]. The majority of criminal justice referrals involve treatment for alcohol or marijuana use. Moreover, 52% of admissions to publicly-funded treatment for marijuana use are referred by the criminal justice system, and this proportion has not changed substantially since at least 1997 [13,14]. Racial breakdowns among treatment admissions are noteworthy. Among persons aged 12 and older, Blacks comprise 19% of all admissions to publicly-funded treatment, and 32% of treatment admissions for marijuana use [13].

Understanding the interplay between race, marijuana use, and criminal justice referrals to treatment is important. Perhaps the need for treatment is greater among Blacks than Whites, hence the disproportionate number of Blacks who receive treatment for marijuana use helps to reduce racially-based health disparities that impact on individuals from low-income backgrounds in particular [15]. Alternatively, perhaps the interplay among race, marijuana, and criminal justice is shaped largely by policies and practices that have framed the War on Drugs in the U.S.

## 2. Race, Drug Offenses, and Criminal Justice

The U.S. drug war has had a profound effect on every stage of the criminal justice system, commencing with arrest. Between 1982 and 2007, arrests for drug possession in the U.S. tripled from an estimated 538,100 in 1982 to over 1.5 million in 2007 [16,17]. Moreover, this increase occurred during an era when arrest rates for serious offenses were declining [18]. About half of all arrests for drug possession involve marijuana [19,20]; in some states, arrests for marijuana possession constitute over 60% of all drug arrests [21]. Black–White disparities in criminal justice processing linked to drugs are noteworthy. In 2010, for example, Blacks were 2.6 times more likely than Whites to be arrested for drug possession in the U.S. (Blacks = 7.0 per 1000 population, Whites = 3.7; figures calculated by the authors from data reported by Snyder, 2012: 23; [22]). State-level data are more telling. In 2013, the American Civil Liberties Union (ACLU) [21] (p. 130, Table A5) launched its investigative report into racial differences in arrests for marijuana possession. Black/White arrest differences were found to disadvantage Blacks in every U.S. state except Hawaii. Alleged links between racial disparities and drug arrests have become so politicized that the *New York Times* published the ACLU data only after the data analyses had been validated by researchers at Stanford University [23].

Race differentials in arrests are inconsistent with data on drug use prevalence that tend to show higher rates of use among Whites or similar rates of use between Whites and Blacks [21,24]. Nguyen and Reuter [19] found that the arrest rate for drug possession among adults was more than three times higher for Blacks compared to Whites, despite relatively equal past-year prevalence and frequency of using marijuana (measured in days). Racial disparities in drug possession arrests are often viewed as a consequence of policing strategies that focus on African-Americans, particularly those who reside in low-income neighborhoods [18,25,26,27]. Additionally, the greater likelihood of outdoor drug transactions among Blacks [20,28] and the limited number of “spatial and social barriers” [18] (p. 192) in low-income neighborhoods means that drug transactions in these areas are more visible to police. In a wider context, the erosion of constitutional protections in the U.S. [29,30,31] has worked to expand police powers in the areas of arrest, search, and seizure.

Large numbers of people arrested for drug possession and other drug-related offenses enter the system of criminal justice, and like all “systems”, one stage of the process impacts on the next, and so on. Racial disparity that disadvantages Blacks tends to accumulate in a series of criminal justice decisions and processes. The disproportionate number of arrests among African-Americans from low-income backgrounds further impacts on Blacks because convictions accrue over time [32] and previous convictions greatly influence the severity of sentencing. Moreover, prosecutors have been found to more often support mandatory minimum sentencing for Black defendants compared to Whites [33]. Discriminatory practices in judicial sentencing have also been documented. Results from a meta-analysis of 76 published and unpublished U.S. studies of race, ethnicity, and sentencing that controlled for offense seriousness and prior criminal record showed that African-Americans were sentenced more severely than Whites in both state and federal courts [34,35]. Although the authors noted that racial disparity in punishment was sometimes minimal, the disparity widened for drug offenses and in federal courts post-1980.

### Criminal Justice Referrals to Community-Based Treatment: Racial Disparities and Net-Widening

Racial disparities have also been observed in the context of referrals to community-based drug treatment in the U.S. These referrals originate from drug courts, probation, other diversionary processes, and to a lesser extent, parole. Among males convicted of drug-related non-violent offenses in California, Blacks were significantly less likely than Whites to be diverted to drug treatment [32,36], although the disparity decreased somewhat when controlling for criminal record [36]. Still, Black/White disparities were more substantial for dispositions that involved diversions to treatment, compared to Black/White differences in length of prison sentences among those who received prison dispositions. The authors suggested that unchecked discretion by court personnel appeared to influence decisions to divert individuals to drug treatment [32].

Although treatment in criminal justice settings is often lacking for people who need it [37], evidence also suggests that criminal justice referrals to treatment can inappropriately “capture” individuals who are not in need of treatment. Referrals are often initiated by probation officers or judges who might lack clinical knowledge about problem drug use and treatment need. Moreover, criminal justice actors’ perceptions of risk are sometimes guided more by the primary charge or conviction than they are by the behaviors of the individual [38]. These factors can widen the diagnostic net and can waste valuable treatment resources. Results from a nationally representative sample of adults in the U.S. showed that large numbers of individuals had engaged in drug treatment services even though they did not meet criteria for substance abuse or dependence [1]. This disconnect was particularly noteworthy among respondents who had been on probation or parole and in substance use treatment during the last 12 months (74% of whom did not meet diagnostic criteria for substance use disorders). In a separate study, DeMatteo and colleagues [39] reported data on weekly urine specimens collected from 284 drug court participants. The authors found that 34% of the samples were drug-negative over a period of 14 weeks. Additionally, approximately two-thirds of the drug-negative group had baseline Addiction Severity Index drug composite scores that significantly predicted the drug-negative outcome over the 14 weeks. The authors concluded that a sizeable number of drug court participants did not appear to show major problems with drug use at baseline, and hence may have been inappropriately sentenced to drug court. The study was limited in that it relied on urinalysis results as an outcome measure, was restricted to 14 weeks, and was based on participants who had been convicted of misdemeanor drug charges (i.e., possession/use of marijuana, possession of drug paraphernalia including needle/syringes, and first offense for driving under the influence). Although additional research is needed, these findings suggest the possibility that the validity of diagnosing substance use problems among criminal justice populations may be limited in some settings. To explore this issue, it may be helpful to take a look at how diagnoses of substance use problems are often made.

## 3. DSM-IV: A Source of Net-Widening for Criminal Justice Populations?

The Diagnostic and Statistical Manual of Mental Disorders (DSM) is often used by clinicians to ascertain the nature and severity of substance use problems among individuals referred to treatment. The DSM is an integral part of diagnosis and treatment planning and is also used for research purposes and financial billing. Diagnosis is critical, and clinicians choose the appropriate diagnosis after assessing individuals and their symptoms. DSM-IV was adopted in 1994, and later revised to DSM-IV-TR (text revisions; herein described as DSM-IV) [40]. The DSM-IV listed symptoms separately for substance dependence and substance abuse. *Dependence* was definitive for individuals who met a minimum of three of seven criteria at any time within a 12-month period: (1) tolerance; (2) desiring or experiencing unsuccessful efforts to reduce substance use; (3) consuming the substance for longer times or in larger amounts than intended; (4) continuing with substance use despite substance-related physical or psychological problems; (5) spending considerable amounts of time trying to obtain, use, or recover from the substance; (6) neglecting or reducing important social, recreational, or occupational activities because of substance use; and (7) withdrawal. A diagnosis of *abuse* was ascertained if recurrent behaviors met one or more of the following four criteria (absent a diagnosis of dependence): (1) failing to meet important obligations relating to home, work, or school; (2) using the substance in situations when it is physically dangerous; (3) experiencing legal problems that are related to substance use; and (4) continuing to use the substance despite substance-related social or interpersonal problems. Under DSM-IV, substance dependence was generally interpreted as being more severe than substance abuse [41]. DSM training manuals featured vignettes that described how diagnoses of substance dependence “take precedence” over diagnoses of substance abuse [42] (p. 140), and individuals who met criteria for both abuse and dependence were generally diagnosed with the latter.

Despite these well-known problems with the DSM-IV as a guide to diagnoses of substance use problems, these criteria were used for substance use diagnoses for nearly 20 years, until the publication of DSM-5 in 2013 [43]. The latter incorporated several changes, some of which are relevant to the present study. First, DSM-IV distinguished between substance “abuse” and “dependence.” The latter showed strong reliability and validity, and correlated highly with substance use problems, however, validity and reliability were considerably weaker for the “abuse” classification [44]. The DSM-5 merged abuse and dependence into one category: substance use disorder (e.g., cannabis use disorder; alcohol use disorder). Second, under DSM-IV, individuals could be diagnosed with substance abuse if they met only one criterion. In a nationally representative survey of the non-institutionalized U.S. population, 63% of current diagnoses for cannabis abuse resulted from one criterion: recurrent substance use in physically dangerous or hazardous situations [45], such as driving a car or operating machinery. Individuals could also be diagnosed with substance abuse if dependence criteria had not been met, and legal problems represented the *only* abuse criterion that an individual had experienced [46,47]. In a study of adolescents—93% of whom were African-Americans—discordance between DSM-IV and DSM-5 classification for cannabis disorders was due solely to the legal criterion [48]. Among non-institutionalized populations, the “legal problems/involvement” criterion was found to be substantially less useful for diagnosing substance use disorders [44] and the symptom was omitted in DSM-5—but only after thousands of individuals had been impacted by it.

As a diagnostic tool, the DSM-IV may have had limited validity when used with criminal justice populations in particular. Surprisingly, there has been little empirical work that has focused on DSM-IV criteria and criminal justice populations, although scholars in the fields of public health, psychiatry, and psychology have noted the limitations of the criterion of legal involvement for diagnoses of substance abuse. To our knowledge, the validity of DSM-IV criteria among criminal justice populations has not been adequately addressed by researchers even though the proliferation of scholarly studies into drug treatment and the criminal justice system has spanned several years. DeMatteo and colleagues [49] (p. 116) noted the potential problems of DSM-IV criteria in their study of drug court participants whom they perceived to be low risk:
“…to receive a diagnosis of substance abuse, a client only needs one symptom recurring in a 12-month period. Legal entanglement is one such symptom, which is not difficult to satisfy for individuals involved in the criminal justice system. This can lead practitioners to the tautological conclusion that anyone arrested for drug possession is, by definition, a drug abuser.”

The authors also reviewed the DSM-IV symptoms that characterized drug dependence. They noted, for example, that involvement in drug-seeking activities was viewed as one symptom of drug dependence, and that these kinds of criteria might be too broad when used with criminal justice populations.

## 4. Aim of the Study

Our main purpose in this study is to examine race (Black/White) differences in diagnoses of cannabis abuse versus dependence. We are particularly interested in whether clients’ race differentiates between diagnoses of cannabis abuse and dependence among individuals referred to treatment by the criminal justice system. To better understand this relationship, we also explore Black/White differences in diagnoses among individuals referred to treatment by healthcare providers, self, and other sources. Our aim is to investigate how race, and to a lesser extent other socio-demographic and drug use variables, impact on diagnoses of cannabis use disorders, specifically, diagnoses of cannabis abuse versus cannabis dependence.

## 5. Materials and Methods

Data for the present study are from the Treatment Episode Data Set (TEDS), and reflect admissions to mostly publicly-funded substance abuse treatment services in the U.S. The unit of analysis is the admission event rather than the individual, and the data are collected annually. We examine treatment admissions for marijuana use during 2011, and we restricted the analysis to outpatient or community-based treatment (excluding inpatient, hospital, and detoxification services). We also limited our analysis to a single state (North Carolina) so that referrals to treatment were more likely to follow the same regulations, processes, and healthcare payment schemes within the context of treatment availability in one state [32] for justification of single-state analyses. At this writing, recreational and medical marijuana remain illegal in North Carolina, similar to other states in the southeastern USA. In North Carolina, treatment referrals from the criminal justice system come largely from courts, probation, or Treatment Accountability for Safer Communities (TASC) which then serves as a liaison between treatment providers and probation officers. Although judges and probation officers refer individuals to treatment and/or related services, they do not administer the DSM. Rather, the DSM is administered by clinicians or counselors who work in services to which individuals are referred after they have been convicted of criminal offenses. These services have contractual arrangements with criminal justice systems and are paid by the State when services are utilized.

Our interest here is in Black/White differences, so we excluded treatment admissions for individuals identified as Native American, Asian and other racial categories in the dataset. We also restricted the sample to DSM-IV diagnoses of either cannabis abuse or cannabis dependence.

### Measures

The dependent variable was diagnosis, coded 1 for cannabis dependence and 0 for cannabis abuse. Referral sources included criminal justice, healthcare professional, self, and other (e.g., schools, employers, community advocates). Race was a binary variable (1 = Black; 0 = White). Other independent and control variables were included because of their potential to affect variations in dependence and abuse diagnoses. Most demographic variables were dichotomous and measured by the presence (coded 1) or absence (coded 0) of the trait. These included female, high school graduate/GED, and homeless at the time of admission. Although our primary interest was on Black/White differences in referrals, some respondents identified as Black or White and also as Hispanic. As the largest ethnic minority in the U.S., individuals of Hispanic ethnicity might also experience disparate treatment. Therefore, we included a variable measuring Hispanic ethnicity (coded 1 for Hispanic, 0 for non-Hispanic). Age was an ordinal variable with 11 categories, hence we treated it as interval. Employment status was categorical from which we created three dummy variables (employed part- or full-time, unemployed, not in labor force). Having other psychiatric problems was coded 1 (yes) and 0 (no). We included three indicators of drug use or drug use history at admission: (1) two or three substances in addition to marijuana (yes = 1; marijuana only = 0); (2) frequency of use during the past 30 days (dummy coded: daily, less than daily, and no use during the past 30 days); and (3) prior treatment (yes = 1; no = 0).

The TEDS data are in the public domain. The Institutional Review Board at Fayetteville State University exempted the study because the data were de-identified at source and the study met federal criteria for exemption.

## 6. Results

We first explore Black/White differences by referral source for North Carolina and compare these results to Black/White differences by referral source for the South-Atlantic States, and the U.S. (see Table 1). Again, the percentages pertain to outpatient admissions for either cannabis abuse or dependence in 2011. Black/White differences in diagnoses of cannabis use disorder (abuse or dependence) are noteworthy, particularly in North Carolina where 69% of Blacks were referred for treatment by the criminal justice system compared to 49% of Whites. In contrast, Whites were about twice as likely as Blacks to be referred to treatment by a healthcare provider (11%; 5%) or via self-referral (27%; 14%). Similar patterns emerged in the South-Atlantic states and for the U.S. as a whole, however, Black/White differences were most pronounced for the state of North Carolina.

### 6.1. Descriptive Information by Referral Source

Sample characteristics by referral source are presented in Table 2. The results pertain to outpatient admissions for cannabis abuse and dependence in North Carolina during 2011. The majority of admissions for cannabis abuse or dependence—60%—were referred by the criminal justice system. Eight percent were referred by a healthcare provider, 20% through self-referral, and 12% through other referral sources. Diagnoses of cannabis abuse differed significantly across referral sources, with the highest proportion of abuse diagnoses among criminal justice referrals (51%). Referrals by healthcare providers, self, and other sources were significantly more likely to result in diagnoses of cannabis dependence, compared to referrals from the criminal justice system which tended to result in diagnoses of cannabis abuse. Race differences across referral sources are also noteworthy. Specifically, Blacks diagnosed with cannabis use disorders were significantly more likely to be referred by the criminal justice system (64%) than they were from other referral sources (range: 36% to 53%). The opposite pattern was observed for Whites, who were significantly more likely to be referred to treatment by sources other than the criminal justice system (36%).

Data in Table 2 also show that although Hispanics were significantly more likely to be referred by healthcare providers (2%) compared to other sources (1%), these differences were not substantial. Females were significantly more likely to be referred by healthcare providers (33%), self (34%) and other sources (36%) than they were by the criminal justice system (18%). Criminal justice referrals were significantly younger (21–24 years) than all other referral sources (25–29 years) and less likely to be homeless (<1%). Across all referral sources, more than half of admissions had a high school diploma or its equivalent, however, the proportion was significantly lower among criminal justice referrals (54%) compared to referrals from healthcare providers (62%), self (60%) and other sources (61%). Criminal justice referrals were significantly more likely to be employed (38%) and more than twice as likely to be employed as individuals referred by healthcare providers (16%) and self (18%).

Past 30-day drug use showed significant differences across referral sources; individuals referred by the criminal justice system were significantly more likely to report no drug use in the past 30 days (41%) and less likely than other individuals to report daily use during the past 30 days (15%). They were significantly more likely to report some drug use during the past 30 days (43%) compared to referrals from healthcare providers (40%), self (39%) and other sources (37%). Thirteen percent of criminal referrals reported using substances other than cannabis. In contrast, over 70% of referrals from healthcare providers, self, or other sources had problems with substances in addition to marijuana. About one-fifth of criminal referrals reported prior treatment for substance use, compared to 44% referred by healthcare providers, 45% who were self-referred, and 37% who were referred by other sources. Having other psychiatric problems was observed in 7% of criminal justice referrals. This proportion was significantly and substantially lower than admissions referred by healthcare providers (52%), self (50%), and other sources (39%). Overall, the results presented in Table 2 suggest very different profiles among treatment admissions across referral sources. Specifically, with treatment admissions from criminal justice sources featuring less severe problems related to substance use.

### 6.2. Multivariate Results

Multivariate results are presented in Table 3. As described previously, DSM-IV diagnosis was treated as a binary dependent variable coded 1 for cannabis dependence and 0 for cannabis abuse. To better understand the effect of race on DSM-IV classification, we estimated four separate models based on sub-samples of referral source: criminal justice, healthcare provider, self, and other sources. Results are derived from logistic regression.

#### 6.2.1. Criminal Justice Referrals

Race emerged as a significant predictor across all models; however, the direction of the coefficient differed for individuals referred by criminal justice sources. Specifically, among criminal justice referrals, Blacks were significantly *less likely* than Whites to be diagnosed with cannabis dependence, controlling for demographic and substance use variables (AOR = 0.89; CI = 0.80–0.98). Stated alternatively, Blacks referred by criminal justice sources were significantly more likely than Whites to be diagnosed with cannabis abuse, rather than dependence. In contrast, Blacks were significantly *more likely* than Whites to be classified with cannabis dependence when referred by healthcare providers (AOR = 1.36; CI = 1.02–1.82), self (AOR = 1.63; CI = 1.35–1.66), or other referral sources (AOR = 1.35; CI = 1.07–1.70). Race was the only demographic variable to reach statistical significance in all four models.

Among criminal justice referrals, four other demographic variables were statistically significant. A diagnosis of cannabis dependence was significantly more likely among individuals who were older (AOR = 1.09; CI = 1.07–1.12), and had less than a high school education (AOR = 0.82; CI = 0.74–0.91), and significantly less likely for individuals who were employed (AOR = 0.81; CI = 0.72–0.89) or not in the labor force (AOR = 0.54; CI = 0.47–0.64), compared to individuals who were unemployed. Results for this model also show that variables relating to substance use were all statistically significant and in the expected direction. Specifically, a diagnosis of cannabis dependence was significantly more likely among individuals who used two or more substances (AOR = 1.92; CI = 1.63–2.27), used daily (AOR = 4.62; CI = 3.93–5.43) and less than daily (AOR = 1.15; CI = 1.04–1.27) compared to no use in the past 30 days, and had a history of prior treatment (AOR = 1.25; CI = 1.10–1.40). Having other psychiatric problems significantly decreased the likelihood of cannabis dependence diagnosis (AOR = 0.63; CI = 0.51–0.79) among admissions referred by criminal justice sources.

#### 6.2.2. Referrals by Healthcare Providers

Results from the second model pertain to the sub-sample of admissions referred by healthcare providers. In addition to the significant effect of race on diagnosis (AOR = 0.73; CI = 0.54–0.97), two other demographic variables emerged as statistically significant. A diagnosis of cannabis dependence was significantly less likely among females compared to males (AOR = 0.72; CI = 0.54–0.97), individuals not in the labor force compared to the unemployed (AOR = 0.63; CI = 0.41–0.96), and individuals using cannabis less than daily (AOR = 0.67; CI = 0.48–0.95) compared to those reporting no use during the past 30 days. The probability of being diagnosed with cannabis dependence was significantly more likely among individuals using substances in addition to marijuana (AOR = 1.71; CI = 1.25–2.36), and those who used marijuana daily during the past 30 days (AOR = 1.50; CI = 1.04–2.17).

#### 6.2.3. Self-Referrals to Marijuana Treatment

Among self-referrals to marijuana treatment, Blacks (AOR = 1.63; CI = 1.35–1.66), individuals with a high school diploma or its equivalent (AOR = 1.38; CI = 1.15–1.67), those presenting as homeless (AOR = 1.67; CI = 1.06–2.62), use of other substances in addition to marijuana (AOR = 1.29; CI = 1.06–1.56), and daily use in comparison to no use during the past 30 days (AOR = 1.71; CI = 1.35–2.16) significantly increased the likelihood of being diagnosed with cannabis dependence. In contrast, using marijuana less than daily decreased the probability of cannabis dependence diagnosis (AOR = 0.70; CI = 0.56–0.87) compared to not using at all during the past 30 days.

#### 6.2.4. Referrals from Other Sources

The final equation provides estimates for referrals from other sources. Blacks (AOR = 1.35; CI = 1.07–1.70), individuals using substances in addition to marijuana (AOR = 1.44; CI = 1.12–1.84), and those who used daily (AOR = 3.54; CI = 2.61–4.80) were more likely to be diagnosed with cannabis dependence. Alternatively, individuals who were employed (AOR = 0.75; CI = 0.57–0.99) and those who were not in the labor force (AOR = 0.28; CI = 0.19–0.41) were less likely to be diagnosed with cannabis dependence compared to individuals who were unemployed.

## 7. Discussion

The primary aim in this study was to determine whether and to what extent race affects diagnosis of cannabis disorders among individuals who are referred to outpatient treatment by the criminal justice system. We examined Black/White differences in diagnosis of cannabis abuse versus cannabis dependence among criminal justice referrals and compared the results to Black/White differences associated with referrals to marijuana treatment from other sources (i.e., health professionals, self, other referral). The results of this study suggest that criminal justice referrals to marijuana treatment might inappropriately have targeted Blacks in particular. Three findings undergird this conclusion.

First, Blacks referred to treatment by the criminal justice system were significantly *less likely* than Whites to be diagnosed with cannabis dependence, controlling for demographic and drug use variables. Stated alternatively, Blacks referred to treatment by the criminal justice system were more likely than Whites to be diagnosed with cannabis abuse. Under DSM-IV, dependence was generally viewed as being more severe than abuse [42]. Thus, the race-disparate diagnoses observed among criminal justice referrals suggest that Blacks may have experienced fewer problems with marijuana use compared to Whites. We suggest the possibility that clinicians’ diagnoses of cannabis abuse were confirmed because criminal justice referrals by definition satisfied one criterion of abuse, i.e., legal problems relating to substance use.

Second, Blacks referred by healthcare providers, self, or other sources were *more likely* than Whites to be diagnosed with cannabis dependence. The multivariate models controlled for various socio-demographic and drug use variables, including the frequency of marijuana use in the past 30 days, polysubstance use, and prior treatment episodes.

Third, the results suggest very different profiles across referral sources. Specifically, individuals referred by the criminal justice system appeared to have used marijuana less frequently in the past 30 days and to have been less likely to use substances other than marijuana, compared to individuals referred by three other sources. Moreover, individuals referred by the criminal justice system were less likely to present with psychiatric problems and more likely to be employed in part- or full-time work compared to individuals referred to marijuana treatment from other sources.

Collectively, these findings suggest that criminal justice referrals were more likely to involve individuals who experienced fewer problems associated with marijuana use, mental health, and work compared to individuals referred from other sources. Whites were admitted to treatment with more serious and qualitatively different symptoms than Blacks. In North Carolina, TASC (Treatment Accountability for Safer Communities) provides a major role in screening individuals for drug treatment, and often serves as an important liaison between treatment providers and probation officers. TASC can refer to treatment when there is “*potential* substance abuse…including drug-related charges” [50] (authors’ emphasis). Indeed, data from 2009 to 2010 show that 50% of TASC clients were convicted of drug offenses [50]. We suggest that although drug-related charges may be indicative of drug use, the offense for which one is charged does not necessarily equate to abuse or dependence. Still, perceptions about “potential substance abuse” might have been shaped partly by convictions for marijuana-related offenses.

Although the data used in the present study cannot be used to explore whether referral and screening for treatment were fueled by racially-based stereotypical assumptions about marijuana use among African-Americans, clearly race-linked stereotypical assumptions have been found to influence other and more severe diagnoses, including conditions diagnosed by trained and licensed clinicians [51,52,53]. Although we do not have access to observational data pertaining to clinical encounters or information about offenses for which individuals were convicted, the results suggest that the profile of criminal justice referrals was distinctively different than the profiles of individuals who were referred by healthcare professionals, self, and other sources.

One possible explanation for these findings is that criminal justice referrals to treatment for marijuana use have worked to extend the War on Drugs to the rehabilitative ideal. The disproportionate number of Blacks who are arrested for marijuana possession might be one factor that has contributed to the large number of criminal justice referrals to treatment for marijuana use. In other words, Blacks may have been more likely than Whites to satisfy the diagnostic criteria for cannabis *abuse* simply because they had a greater likelihood than Whites to come into contact with the criminal justice system for marijuana offenses. This kind of differential works to extend racially-based disparities that occur at earlier stages of the criminal justice system.

Additionally, the results from the present study suggest that referrals to marijuana treatment from criminal justice sources may have widened the diagnostic net by involving people in marijuana treatment who may not have needed treatment at all. Among criminal justice referrals, the over-reach of the diagnostic net appears to have impacted Blacks more so than Whites. Newer nets tend to shift social control from one system to another [54], however, newer nets can also result in an expansion of social control that is not shifted but shared, here between criminal justice and drug treatment services [55]. Surveillance and monitoring become more intensive so that sentences that incorporate quasi-coerced drug treatment can be difficult to serve. Violations of probation, for example, are common. The data for the present study were collected in 2011, during which 50% of North Carolina prison admissions involved probation violations [56].

The findings suggest a mismatch between treatment need and provision for some individuals referred by the criminal justice system. If the gap between treatment need and availability is extensive among criminal justice populations [6,7], it might be more effective to utilize the limited treatment provision for individuals with greater need than the individuals diagnosed with cannabis abuse in the present study.

As noted earlier, the DSM-5 was adopted in 2013 and replaced cannabis abuse and dependence with one classification, cannabis use disorders (CUD), further defined in terms of the severity of symptoms (mild, moderate, or severe). A diagnosis of CUD is met if individuals meet two of 11 symptoms over a 12-month period. The legal involvement criterion has been officially removed from diagnostic consideration. It is not yet known how this change will affect diagnoses of substance use problems among criminal justice populations. Future research should investigate the severity of CUD symptoms (mild, moderate, or severe) by clients’ race and across referral sources to ascertain whether racial disparities in substance use diagnoses have diminished. Although DSM-IV is not the most current edition, its diagnostic consequences have continued to impact people’s lives. Despite the release of DSM-5, a survey conducted in 2014 of more than 6000 mental health care practitioners including psychiatrists, psychologists and family practitioners, found that 55% still were not using the DSM-5. Further, approximately one-fifth reported that the latest edition was not relevant to their practice [57]. Therefore, it is imperative to consider how individual lives have been affected by the gap between the clinical application of the two editions.

## 8. Limitations of the Study

This study is not without limitations. First, TEDS admissions data derive largely from publicly funded treatment services. We do not have data on the number of for-profit treatment services in North Carolina, thus we are uncertain about the number of admissions to private treatment services within the state. Including those data might have yielded different results. Second, we had no information on the specific DSM-IV symptoms that were used to make diagnostic decisions or the number of criteria that were met per admission. We also acknowledge that clinicians can utilize various assessment tools in addition to the DSM. Our inclusion of control variables that measured recent drug use, polysubstance use, psychiatric problems, and prior treatment episodes helped to address this limitation.

## 9. Conclusions

Despite limitations of this study, its conclusions are clear: Race matters in the diagnosis of cannabis disorders. Specifically, when referred from the criminal justice system, African American individuals more than Whites tend to have less serious problems with marijuana use, implying reduced need for treatment. This disparity is revealed through the DSM-IV, which allowed for diagnoses of cannabis abuse based on legal problems, e.g., involvement with the criminal justice system. One consequence of this disparity is net widening that places more African American persons under greater layers of social control. Another is the development of a treatment gap, where scarce resources are given to those with the least need, leaving those with more serious substance use problems wanting. With the recent creation of the diagnostic category of cannabis use disorders (present in the DSM-5), the legal criterion has been abandoned. However, it remains to be seen whether and how this change will affect racial disparities in criminal justice referrals to treatment for marijuana use, diagnoses for cannabis use disorders, or the associated consequences. National treatment admission data using DSM V should be available for secondary analysis by December 2017.

## Figures and Tables

**Table 1 behavsci-06-00021-t001:** Admissions to outpatient treatment with DSM diagnosis of either cannabis dependence or cannabis abuse, by race (blacks and whites) and referral source, 2011.

Referral Source	Black	White
North Carolina (*N* = 12,112)		
Criminal Justice	69%	49%
Healthcare provider	5%	11%
Self	14%	27%
Other referral	12%	13%
South Atlantic states ^a^ (*N* = 41,170)		
Criminal Justice	57%	48%
Healthcare provider	3%	5%
Self	11%	17%
Other referral	27%	30%
US (*N* = 109,287):		
Criminal Justice	61%	55%
Healthcare provider	3%	4%
Self	12%	17%
Other referral	25%	24%

^a^ South Atlantic states are a Census Division and include DE, DC, FL, GA, MD, NC, SC, VA, and WV.

**Table 2 behavsci-06-00021-t002:** Characteristics of treatment admissions to outpatient services for cannabis dependence or cannabis abuse by referral source, North Carolina (2011).

Treatment Admission Characteristics by Referral Source						
Treatment Admission Characteristics	All	Criminal Justice	Healthcare Provider	Self	Other Source	Significance
*Referral source*						
Criminal justice	60%	---	---	---	---	---
Healthcare provider	8%	---	---	---	---	---
Self	20%	---	---	---	---	---
Other	12%	---	---	---	---	---
*DSM-IV cannabis*						
Abuse	46%	51%	42%	37%	37%	*p* ≤ 0.001
Dependence	54%	49%	58%	63%	63%	*p* ≤ 0.001
*Race*						
Black	56%	64%	36%	40%	53%	*p* ≤ 0.001
White	44%	36%	64%	60%	47%	*p* ≤ 0.001
*Hispanic*	1%	1%	2%	1%	1%	*p* ≤ 0.05
*Female*	24%	18%	33%	34%	36%	*p* ≤ 0.001
*Age* (mean category, in years)	25–29	21–24	25–29	25–29	25–29	*p* ≤ 0.001
*Homeless*	3%	<1%	8%	5%	5%	*p* ≤ 0.001
*High school graduate/GED*	57%	54%	62%	60%	61%	*p* < 0.001
*Employment*						
Part- or full-time	30%	38%	16%	18%	22%	*p* ≤ 0.001
Unemployed	57%	49%	73%	67%	67%	*p* ≤ 0.001
Not in labor force	13%	13%	11%	15%	11%	*p* ≤ 0.01
*Past 30 days*						
No use, past 30 days	35%	41%	24%	22%	30%	*p* ≤ 0.001
Less than daily use, past 30 days	41%	43%	40%	39%	37%	*p* ≤ 0.001
Daily use, past 30 days	24%	15%	36%	39%	33%	*p* ≤ 0.001
*Two or more substances*	37%	13%	76%	71%	72%	*p* ≤ 0.001
*Prior treatment*	30%	21%	44%	45%	37%	*p* ≤ 0.001
*Other psychiatric problems*	23%	7%	52%	50%	39%	*p* ≤ 0.001
N	12,116	7304	937	2381	1494	

**Table 3 behavsci-06-00021-t003:** Logistic regression estimates for cannabis dependence, by referral source among outpatient admissions, North Carolina, 2011.

Referral Source																
Treatment Admission Characteristics	Criminal Justice				Health Care Provider				Self				Other			
	b	SE_(b)_	OR	95% CI	b	SE_(b)_	OR	95% CI	b	SE_(b)_	OR	95% CI	b	SE_(b)_	OR	95% CI
*Black*	−0.12 *	0.05	0.89	(0.80–0.98)	0.31 *	0.15	1.36	(1.02–1.82)	0.49 ***	0.10	1.63	(1.35–1.66)	0.30 **	0.12	1.35	(1.07–1.70)
*Age*	0.09 ***	0.01	1.09	(1.07–1.12)	0.01	0.03	1.01	(0.94–1.06)	0.04	0.02	1.04	(0.99–1.08)	−0.01	0.03	0.99	(0.94–1.05)
*Female*	−0.12	0.07	0.89	(0.78–1.01)	−0.33 *	0.15	0.72	(0.54–0.97)	−0.04	0.09	0.96	(0.80–1.16)	0.17	0.13	1.19	(0.93–1.51)
*Hispanic*	−0.22	0.29	0.80	(0.45–1.43)	1.32	0.68	3.74	(0.98–14.25)	0.48	0.41	1.61	(0.72–3.59)	−0.48	0.50	0.62	(0.23–1.64)
*HS grad/GED*	−0.20 ***	0.05	0.82	(0.74–0.91)	0.04	0.15	1.04	(0.78–1.38)	0.33 ***	0.10	1.38	(1.15–1.67)	−0.04	0.13	0.96	(0.76–1.23)
*Employment*																
Employed	−0.22 ***	0.05	0.81	(0.72–0.89)	0.09	0.20	1.09	(0.75–1.61)	−0.01	0.12	0.99	(0.79–1.25)	−0.29 *	0.14	0.75	(0.57–0.99)
Not in labor force	−0.61 ***	0.08	0.54	(0.47–0.64)	−0.47 *	0.22	0.63	(0.41–0.96)	−0.18	0.13	0.83	(0.64–1.08)	−1.29 ***	0.20	0.28	(0.19–0.41)
Unemployed	1.00				1.00				1.00				1.00			
*Homeless*	0.33	0.39	1.39	(0.65–3.01)	−0.25	0.27	0.78	(0.46–1.31)	0.51 *	0.23	1.67	(1.06–2.62)	0.37	0.29	1.45	(0.82–2.58)
*2–3 substances*	0.65 ***	0.09	1.92	(1.63–2.27)	0.54 ***	0.16	1.71	(1.25–2.36)	0.25 *	0.10	1.29	(1.06–1.56)	0.36 **	0.13	1.44	(1.12–1.84)
*Past 30 days*																
Daily	1.53 ***	0.08	4.62	(3.93–5.43)	0.41 *	0.19	1.50	(1.04–2.17)	0.54 ***	0.12	1.71	(1.35–2.16)	1.26 ***	0.16	3.54	(2.61–4.80)
<Daily	0.14 **	0.05	1.15	(1.04–1.27)	−0.39 *	0.18	0.67	(0.48–0.95)	−0.36 **	0.12	0.70	(0.56–0.87)	0.03	0.13	1.03	(0.79–1.34)
No use	1.00				1.00				1.00				1.00			
*Prior treatment*	0.22 ***	0.06	1.25	(1.10–1.40)	−0.13	0.14	0.88	(0.67–1.16)	−0.05	0.09	0.95	(0.80–1.14)	0.11	0.12	1.12	(0.88–1.42)
*Other psychiatric problems*	−0.46 ***	0.12	0.63	(0.51–0.79)	0.23	0.14	1.26	(0.95–1.67)	−0.10	0.09	0.91	(0.76–1.08)	0.15	0.12	1.16	(0.91–1.48)
Constant	−0.55				−0.10				−0.23				−0.15			
Nagelkerke R^2^	0.12				0.09				0.09				0.17			
Model χ^2^ (df = 13)	663.68				60.23				157.32				197.24			
N	7293				919				2349				1487			

* *p* ≤ 0.05; ** *p* ≤ 0.01; *** *p* ≤ 0.001.

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
