# Peer review of "Black–White Disparities in Criminal Justice Referrals to Drug Treatment: Addressing Treatment Need or Expanding the Diagnostic Net?"

_behavsci, 2016, doi:10.3390/bs6040021_

Round 1

Reviewer 1 Report

Brief summary

This article highlights racial disparities in referrals to drug treatment from the criminal justice system. The introductory sections outline the various components of the criminal justice system wherein racial disparities have been observed, from search and arrest to sentencing. Through secondary data analysis of treatment data for the state of North Carolina (2011), the authors compare diagnoses of cannabis use disorder (abuse and dependence as measured by DSM-IV) across a range of referral sources including the criminal justice system, healthcare providers, self and other sources. Findings highlight race as significant in relation to type of diagnosis, with Blacks less likely to be diagnosed with cannabis dependence when referred through the criminal justice system, but more likely to be diagnosed with cannabis dependence when referred by healthcare providers, self or other sources. The authors suggest that use of DSM-IV and diagnoses of cannabis abuse based on one criterion of experiencing legal problems may have resulted in referrals from the criminal justice system capturing a cohort that experience fewer problems with marijuana use in comparison with other referral sources. It is proposed that Blacks are over-represented in marijuana treatment because they are more likely than Whites to come into contact with the CJS. The authors suggest that treatment referrals for those diagnosed with cannabis abuse further widens the gap between treatment need and availability among those in contact with the criminal justice system.

Broad comments

This paper provides an interesting, convincing and well-structured argument to support existing research on racial disparities observed in the criminal justice system, making an important and valuable contribution to the literature. My comments are as follows:

Some clarity around the role and timing re: administration of the DSM-IV in the process of referral to treatment might be helpful. Line 141 states, “referrals are often initiated by probation officers or judges who might lack clinical knowledge about problem drug use and treatment need.” Are referrals from the criminal justice system founded on diagnoses from DSM-IV or is this administered upon admission to treatment post-referral? In cases where treatment is quasi-coerced and mandatory to avoid prison, if an individual is then diagnosed with neither abuse nor dependence (under newer DSM-5), do they still have to complete treatment as part of CJS requirements? I am trying to ascertain whether the referral process itself is the problem and therefore the diagnosis of abuse versus dependence is used here as a retrospective indicator of treatment need. If so, I think this could be made clearer.

The paper might benefit from brief discussion of decriminalization of marijuana in some US states and the position of North Carolina on this issue. 

The paper would be strengthened with a conclusion that draws together the strong take home messages or subheadings used in the Discussion section to make it easier to digest.

Specific comments

Line 88 – “Black/white arrest differences were found to disadvantage Blacks in every US state except Hawaii.” Any suggestions as to why Hawaii is unique in this respect?

Line 93-95: Prevalence data compares rates between 1991-1992 and 2001-2002. Does more recent comparative data show similar trends?

Line 210: Would it be useful to include example often provided, e.g. driving/operating machinery?

Line 353: Perhaps a sentence justifying the inclusion of Hispanic ethnicity.

Line 422: “decreased the likelihood of cannabis dependence” – insert “diagnosis”.

Line 496: New paragraph beginning “Collectively…”

Author Response

Word file uploaded.  Many thanks.

Reviewer 2 Report

1)    The research question is about the “relationship between race and diagnosis of cannabis use disorder (dependence or abuse) among referrals to community-based treatment.” The authors are using the DSM IV criteria to distinguish between abuse and dependence. However, in 2013 (three years), the DSM merged these categories.  The authors fail to provide an explanation why DSM changed its categories and why the authors base their study on outdated DSM criteria. The authors also state “Despite these well-known problems with the DSM-IV as a guide to diagnoses of substance use problems,…” Were these problems the reason why the DSM changed its categories? If so, what are the implications of using the prior problematic categories in this study?

2)    In addition, there are problems noted by the author in using the DSM IV criteria with CJ populations. These problems seem to be significant. How does that impact the results of this study? If the study purpose depends on a valid differentiation between abuse and dependence, and the criteria used are not valid the how can the study produce valid results? The authors need to justify their decision and explain impacts on their results.

3)    Why did the authors not use the DSM V to assess referral differences between Whites and Blacks? It’s all in one category, but the question could still be assessed. The authors state the following research question in the Discussion section: “To what extent does race affect diagnosis of cannabis disorders among individuals who are referred to outpatient treatment by the criminal justice system?” Why not use the DSM V to assess this question?

Author Response

Word file uploaded.  Many thanks.

Round 2

Reviewer 2 Report

No comments